

# Global whole-rock geochemical database compilation

Matthew Gard[1], Derrick Hasterok[1,2], and Jacqueline Halpin[3]

[1]Department of Earth Sciences, University of Adelaide, North Terrace, SA, 5005, Australia
[2]Centre for Tectonics Research and Exploration (TRaX), University of Adelaide, North Terrace, SA, 5005, Australia
[3]Institute for Marine and Antarctic Studies (IMAS), University of Tasmania, Hobart, Tasmania, 7001, Australia

**Correspondence:** Matthew Gard (matthew.gard@adelaide.edu.au)

**Abstract.** Dissemination and collation of geochemical data are critical to promote rapid, creative and accurate research and place new results in an appropriate global context. To this end, we have assembled a global whole-rock geochemical database, with other associated sample information and properties, sourced from various existing databases and supplemented with numerous individual publications and corrections. Currently the database stands at 1,023,490 samples with varying amounts

of associated information including major and trace element concentrations, isotopic ratios, and location data. The distribution both spatially and temporally is quite heterogeneous, however temporal distributions are enhanced over some previous database compilations, particularly in terms of ages older than ∼1000 Ma. Also included are a wide range of computed geochemical indices, physical property estimates and naming schema on a major element normalized version of the geochemical data for quick reference. This compilation will be useful for geochemical studies requiring extensive data sets, in particular those

wishing to investigate secular temporal trends. The addition of physical properties, estimated by sample chemistry, represents a unique contribution to otherwise similar geochemical databases. The data is published in .csv format for the purposes of simple distribution, but exists in a format acceptable for database management systems (e.g. SQL). One can either manipulate this data using conventional analysis tools such as MATLAB®, Microsoft® Excel, or R, or upload to a relational database management system for easy querying and management of the data as unique keys already exist. This data set will continue to grow, and we

encourage readers to contact us or other database compilations contained within about any data that is yet to be included. The data files described in this paper are available at https://doi.org/10.5281/zenodo.2592823 (Gard et al., 2019).

## 1   Introduction

Geochemical analyses in conjunction with other temporal, spatial, and physical property data from rock samples have been

vital sources of information for understanding the Earth and investigating local, and global geodynamic histories (e.g. Keller and Schoene, 2018). Effective collection, collation and dissemination of this type of data is critical to promote rapid, creative and accurate research. Every year, the amount of data recorded globally increases dramatically, often dispersed among many hundreds of individual publications. Since the 1960's and 70's, broad element suites have been rapidly accumulated due to





the commercial availability of methods such as x-ray fluorescence (XRF) and inductively coupled plasma mass spectrometry (ICP-MS), and thus modern publications have swiftly expanded our cumulative global data records. However due to the rate of new publications, in conjunction with significant partitioning of this data between different journals, countries, authors etc. this data is not always easy to find and can be incredibly time consuming to collate. It is pertinent that this information be readily available for future studies as all benefit from taking advantage of the full suite of data available to produce more robust models and constrained analyses.

Geochemical compilations have been used in a range of studies such as examining crustal magma reservoirs (e.g. Carbotte et al., 2013), proposing changes in mantle dynamics (e.g. Iwamori and Nakamura, 2015), to look at regional and global tectonic histories (e.g. Keller and Schoene, 2018), and examine the connections between life and the solid Earth (e.g. Cox et al., 2018). Not only does this information have implications for scientific community, but also for issues such as environmental management, land use, and minerals resources development.

In this paper we present a global whole-rock geochemical database compilation consisting of modified whole-rock subsets from existing database compilations, in conjunction with supplementation from individual publications not yet included in these other collections. Additionally, we have generated naming schema, various geochemical indices, and other physical property estimates for a range of the data contained within.

## 2 Existing Initiatives

Many existing initiatives have worked to construct and maintain data compilations with great success, but often restrict themselves to certain tectonic environments or regimes, regions, or rock types.

EarthChem (https://www.earthchem.org/) is currently the most notable 'go-to' geochemical data repository for general use. It consists of many federated databases such as PetDB, NAVDAT, SedDB, the USGS National Geochemical Database and GEOROC, as well as individually submitted publications. The constituent databases are mostly more specialized compilations, for example;

- The North American Volcanic and Intrusive Rock Database (NAVDAT) has existed since 2002 and is primarily aimed at geochemical and isotopic data from Mesozoic and younger igneous samples of western North America (Walker et al., 2006). (http://www.navdat.org/)

- The Petrological Database of the Ocean Floor (PetDB) is the premier geochemical compilation suite for the igneous and metamorphic hosted data from mid-ocean ridges, back-arc basins, sea-mounts, oceanic crust and ophiolites (https://www.earthchem.org/petdb).

- Geochemistry of Rocks of the Oceans and Continents (GEOROC) is a more holistic compilation effort of chemical, isotope, and other data for igneous samples, including whole-rock, glass, minerals and inclusion analyses and metadata (http://georoc.mpch-mainz.gwdg.de).





- SedDB focuses on sedimentary samples, primarily from marine sediment cores. It has been static since 2014, and includes information such as major and trace element concentrations, isotopic ratios, and organic and inorganic components. (http://www.earthchem.org/seddb).

- MetPetDB is a database for metamorphic petrology, in a similar vein to PetDB and SedDB. This database also hosts large swathes of images collected through various methods such as x-ray maps and photomicrographs, although this information is not used in this study (http://metpetdb.com/).

- The USGS National Geochemical Database archives geochemical information and its associated metadata from USGS studies and made available online (https://www.usgs.gov/energy-and-minerals/mineral-resources-program/science/national-geochemical-database).

Many other government initiatives and national databases exist, with notable examples including PETROCH from the Ontario Geological Survey (Haus and Pauk, 2010), New Zealand's national rock database (Petlab) (Strong et al., 2016), Australia's national whole-rock geochemical database (OZCHEM) (Champion et al., 2016), the Finnish lithogeochemical rock geochemistry database (RGDB) (Rasilainen et al., 2007), the Newfoundland and Labrador Geoscience Atlas (Newfoundland and Labrador Geological Survey, 2010), and the basement rock geochemical database of Japanese islands (DODAI) (Haraguchi et al., 2018).

While all of these are generally exceptional enterprises, we personally ran into issues within our own research. Some examples included databases being deficient in aged data (1000 Ma+), or lacking many recent publications. Issues with existing data within these databases was also evident; we found many samples missing information available in the original individual publications. It was quite common for age resolutions to be significantly larger than the values quoted within the paper itself, on the order of hundreds of millions of years in some cases, or not included at all because they were not found in a table but within the text itself.

Thus, we seek to produce a global whole-rock geochemical database incorporating samples from previous databases as necessary, and supplementing significantly from other, often recent, publications. Computed properties, naming schemes, and various geochemical indices have also been calculated where the data permits. Smaller subsets of previous versions of this database have already been utilised for studies of heat production and phosphorus content (Hasterok and Webb, 2017; Hasterok et al., 2018; Cox et al., 2018; Gard et al., in review; Hasterok et al., in review), and this publication represents the totality of geochemical information gathered. As an ongoing process we have corrected some errors or omissions from previous databases as we have come across them, but have not undergone a systematic effort to quality check the prior compilations. We intend to continue updating the database both in additional entries and in further clean up when necessary.

## 3 Database aggregation and structure

While other database structures are incredibly efficient, some of the intricacies of the systems make it difficult to utilise the information contained within. We personally had issues when seeking estimated ages of rock samples. For studies which



examine temporal variations of chemistry or physical properties an accurate and precise age is required. Under some of the present data management schemes for online databases it may be difficult to recover the desired data. Crystallization dates for older samples are often determined by U-Pb or Pb-Pb measurements from a suite of zircons. For a given sample, the individual zircon ages may be contained within the database, and stored under mineral analyses. However, a search for rock

chemistry may return an age (often a geologic timescale division), but not a precise date. To get the date one would have to also download the individual mineral analyses, conduct an age analysis on a concordia diagram (or similar), determine whether each individual analysis was valid, and then associate the result with the bulk chemistry. This process can be tedious and may be intractable. Had the estimated crystallization date been attributed to the sample directly as often reported in the original study, much of this process could be short cut. Our database seeks to do just that, by attributing an estimated crystallization age

to the sample as provided in the original reference at the point of data entry. This also allows us to include estimated dates for the same unit or formation. As a result the database presented here allows for a higher density of temporal sampling than other compilations.

We have chosen a mixed flat file and relational database structure for simpler distribution. Codd (1970) was the first to propose a relational model for database management. A relational structure organises data into multiple tables, with a unique

key identifying each row of the sub-tables. These unique keys are used to link to other sub-tables. The main advantages of a relational database over a flat file format are that data is uniquely stored just once, eliminating data duplication, as well as performance increases due to greater memory efficiency and easy filtering and rapid queries.

Rather than utilize an entirely relational database format, we have adopted some flat file formats for the sub tables as to reduce the number of total tables to an amount more manageable for someone unfamiliar with SQL database structure. This

format raises storage memory due to data duplication in certain fields (e.g. repetition of certain string contents across multiple samples, such as rock name). However, we believe this is a reasonable trade off for an easier to utilize structure for distribution, and makes using this data for someone unfamiliar with SQL simpler. Ideally we would host a purely relational database structure online and be accessed via queries similar to the EarthChem Portal, but this is yet to be done.

We utilise PostgreSQL as the relational database management system (RDMS) to update and administer the database. Post-

25 greSQL containins many built in features and useful addons including the geospatial database extender PostGIS which we utilise, has a large open source community and runs on all major operating systems.

Python in conjunction with a PostgreSQL database adapter Psycopg are used to import new data efficiently. Data is copied into a .csv template directly from publications to reduce any chance of transcribing errors, and dynamically uploaded to a temporary table in PostgreSQL. From here, the desired columns are automatically partitioned up and added to the database in

their respective sub-tables. We iterate through a folder of new publications in this way, and are able to add data rapidly as a result.

The database consists of 10 tables: trace elements, major elements, isotope ratios, sample information, rock group/origin/facies triplets, age information, reference information, methods, country, and computed properties. The inter-connectivity of these tables is depicted in Figure 1, with tables linked via their respective id keys. A description of each of these tables is included in

Table 1, and column names that require further details as well as computed property methods are detailed in Table 3. Individual



subtables have been output as csv files for use. We suggest inserting these into a RDBMS for efficient queries and extraction of desired data. However, we have exported these in csv format in case people not familiar with database systems wish to work with them in other programs such as Microsoft®Excel, MATLAB® or R. While technically inefficient, the largest sub-table currently stands at only 200 Mb, which we believe to be an acceptable size for data manipulation.

Many samples include multiple analyses. These can vary from separate trace and major measurements with no overlap, to duplicate element analyses using different methods. In the case of some subsets of this data we have chosen to merge these multiple analyses into a singular entry in the database. This methodology has both benefits and drawbacks. While it reduces the difficulty in selecting individual samples analyses, it means that lower resolution geochemical methods are sometimes averaged with higher precision ones. In the future we hope to prioritise these higher precision methods where applicable (e.g. ICP-MS

for many trace elements over XRF). Using a singular entry is simpler for many interdisciplinary scientists who don't wish to be slowed down by the complexity of managing duplicate samples and split analyses. We have generally kept track of this with the method field; where merging has occurred and both methods are known, we have concatenated the method in most cases.

## 4   Data statistics

### 4.1   Raw data

The largest existing database contributions to this database are listed in Table 2. Individual publication supplementation includes both new additions we have found online, as well as clean up of entries previously entered into existing databases. The subsets of previous data sets do not represent the entire collections for many of these programs as we have done pre-filtering to remove non-whole rock data, or issues with accessing the entire data sets online.

Figure 2 denotes histograms of the various major, trace and isotope analyses within the database. The majority of isotope

data was recently sourced from the GEOROC database. Major element analyses in general dwarf the amount of trace element measurements recorded in terms of consistency which is unsurprising.

The samples are distributed reasonably well around the globe. However, there is a noticeable dominance of samples sourced from North America, and in a smaller way from Canada, Australia and New Zealand (Figure 3). The United States tops of the list with 335,266 samples, including those from their non-contiguous states. The African continent suffers the most from lack

of data with regards to the rest of the globe (Figure 3).

Age here is indicated as being an assumed crystallization age. Age distributions unsurprisingly show a significant dominance for very recent samples ($<50$ Ma), due largely to the oceanic subset (Figure 4b). Excluding major time-period associated ages (e.g. Paleoproterozoic age range of 2,500–1,600 Ma as the max and min age of a sample), there are 361,815 samples with mean crystallisation age values. Of these, 282,375 have age uncertainty estimates and observing the cumulative distribution

function of these values indicates that $\sim 99\%$ of the age uncertainties fall below $\sim 150$ Ma (Figure 4a).

Rock group and rock origin are described in Table 3. There is a clear dominance towards igneous samples, making up 73.80% of the data with known rock group information (Figure 5). About 99% of these igneous samples have a distinction noted as volcanic or plutonic in the rock origin field, with just over two thirds of these being volcanic. Sedimentary samples are the next





most common rock group, however the vast majority of these have no classification in rock origin, and we aim to improve this in future updates. Finally metamorphic rocks have $\sim 43\%$ of the samples with rock origin classifications. Meta-sedimentary origin is slightly more common than meta-igneous, however meta-igneous includes two further subdivisions of meta-volcanic and meta-plutonic where known.

## 4.2 Computed properties

We compute a number of properties and naming schema for a significant subset of the database, a new addition over many previous database compilations. This includes heat production, density and p-wave velocity estimates, as well as various geochemical indices and descriptors such as modified TAS, QAPF and SIA classifications. A full list of referenced methods and computed columns are given in Table 3. Where computed values require major element concentrations, these properties and values have been calculated based on an LOI free major element normalised version of the database i.e. major element totals are normalised to 100, while preserving the relative proportions of each individual elements contribution to the total. This normalisation occurs only on samples with major element totals between 85 and 120 wt.%. Totals lying outside this range are ignored, and properties requiring these values are not computed. The exact value of normalisation for each sample is recorded in the computed table, within the norm_factor field.

### 4.2.1 Naming schema - rock_type

Nomenclature varies significantly within geology and unsurprisingly rock names within the database differ wildly as a result. Different properties such as texture, mineralogical assemblages, grain sizes, thermodynamic histories, and chemistry make up the majority of the basis for the various naming conventions utilised throughout, interspersed with author assumptions and/or inaccuracies. Thus, we sought a robust and consistent chemical classification scheme to assign rock names to the various samples of the database. This chemical basis classification scheme is stored in the computed table, within the rock_type field.

Differing naming work flows are applied to (meta-)igneous, and (meta-)sedimentary samples. For igneous, meta-igneous, and unknown protolith origin metamorphic samples, we use a total alkali-silica (TAS) schema (Middlemost, 1994) modified to include additional fields for further classification of high-Mg volcanics (Le Bas and Streckeisen, 1991). See Figure 6c and d for a partial visual description of the process. Furthermore we classify igneous rocks as carbonatites when the CO2 concentration exceeds 20 wt.%. These entries are assigned either the plutonic or volcanic equivalent rock names depending if the sample is known to be of plutonic or volcanic origin.

For sedimentary and meta-sedimentary rocks, we first separate out carbonates and soils using ternary plot divisions of $SiO_2$, $Al_2O_3 + Fe_2O_3$, and $CaO + MgO$ (Mason, 1952; Turekian, 1969). Additionally, we further partition clasic sediments using the SedClass™classification method from Herron (1988). Quartzites are identified separately where $SiO_2$ exceeds 0.9 in the ternary system. See Hasterok et al. (2018) for further discussion.

A break down of the classification distributions are included in Figure 6a and b. Sub-alkalic basalt/gabbro is a significantly large contribution to the volcanic samples, unsurprisingly due to the extent of samples of oceanic nature.





### 4.2.2 Computed physical properties

Physical properties for rock types used in numerical models are often based on averages based on limited samples from individual publications. This database provides an opportunity for in-depth analysis of physical properties of rock types with specified chemistry that can be used to improve geodynamic models.

Figure 7a, b and c denote some property estimates calculated from the normalised analyses. Estimates of density and p-wave velocities are based on the methods contained within Hasterok et al. (2018), albeit with a larger data set provided here. By utilising the density estimate we can also compute heat production estimates by employing the relationship from Rybach (1988). The multi-modal nature of the density and p-wave velocity estimates are driven largely by the dichotomy between mafic and felsic samples within the database. Heat production estimates however are resolved by a smoother distribution in

log-space. Density estimates peak at $\sim$2680 and $\sim$2946 $\mathrm{kg\,m^{-3}}$ due to mafic and felsic sample medians respectively, and p-wave velocity estimates depict maximums at 6.183 and 7.135 $\mathrm{m\,s^{-1}}$. Heat production has a median value of $\sim$1.009 $\mathrm{\mu W\,m^{-3}}$, with first and third quartiles (25th and 75th percentiles) of 0.3886 and 2.199 $\mathrm{\mu W\,m^{-3}}$ respectively.

## 5 Improvements and future developments

### 5.1 Bibliographic information

Due to a high variety of sources and database formats, merging bibliographic information proved difficult. For individual publications and adjustments made manually, we have collated bibliographic information in higher detail. For other inherited bibliographic information from external databases, the exact format can vary. These details are contained within the reference .csv and linked to each sample through the ref_id as seen in Figure 1.

### 5.2 Ownership and accuracy

Although every effort is made to ensure accuracy, there are undoubtedly some errors, either inherited or introduced. We make no claims to the accuracy of database entries or reference information. It is up to the user to validate subsets for their own analyses, and ideally contact the original authors, previous database compilation sources, or ourselves to correct errors where they exist. We make no claim on ownership of this data; when utilising this database additionally cite the original authors and data sources.

## 25  6 Bibliography information file

Where DOI exists or where we have manually cleaned up individual publications, we have attached a bibtex file of the entries, containing further information over the reference .csv file. We hope to expand this .bib file as we continue to clean up the reference lists and make adjustments to other compilations. Many do not have this information however as we have inherited

many database reference lists, and for those which don't, the information required to find the sample are included in reference table to the best of our ability.

## 7 Future Work

We have published portions of the database in the course of prior studies and will continue to expand this data set for our own research purposes. Small individual corrections have occurred incrementally with every version, and unfortunately we did not keep records of this improvements. Going forward, we plan to include a record of these corrections and forward them to the other database compilations as needed. We hope to work with existing compilation authors in the future to assist with new additions as well. This version of the database may be of use for these database initiatives to supplement their own records.

Utilising this database we are working on methods for predicting protoliths of metamorphic rocks, as over 57% of the samples lack that information (Figure 5). We are also making progress on a geologic provinces map that captures tectonic terranes. These projects are being done separately, but may be utilised in conjunction with this database in future updates. An associated set of software that can be used in MATLAB® to explore the database, including many of the individual methods cited above for the computed properties subtable will be released some time in the future.

## 8 Data availability

The BIB file and CSV tables of this dataset are available on Zenodo: https://doi.org/10.5281/zenodo.2592823 (Gard et al., 2019)

*Author contributions.* M. Gard and D. Hasterok worked on the processing codes and computed property estimates, as well as collation of data sources. M. Gard organised the database structure and framework codes, and prepared the manuscript with contributions from all co-authors. J. A. Halpin collated the Antarctic geochemical set.

*Competing interests.* The authors declare that they have no conflict of interest.

*Acknowledgements.* We thank Bärbel Sarbas for supplying the GEOROC database in its entirety. We would also like to thank the following individuals for providing data sets and/or personal compilations: D. Champion (GA) D. Claeson (SGU), T. Slagstad (NGU), Lorella Francalanci (UNIFI), Yuri Martynov (FEGI-RAS), Takeshi Hanyu (JAMSTEC), J. Clemens (SUN), H. Furness (UIB), A. Burton-Johnson (BAS) and M. Elburg (UJ). Peter Johnson provided a collection of papers with data for the Arabian-Nubian Shield. M. Gard is supported by Australian Government Research Training Program Scholarship. This research was supported partially by the Australian Government through the Australian Research Council's Discovery Projects funding scheme (project DP180104074). J. A. Halpin was supported under





Australian Research Council's Special Research Initiative for Antarctic Gateway Partnership SR140300001. The views expressed herein are those of the authors and are not necessarily those of the Australian Government or Australian Research Council.



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





**Figure 1.** Database relational structure. Sub-tables are linked through foreign id keys. Ambiguous field names are described in detail in the supplemental material.





**Figure 2.** Histogram of analyses. a) Trace elements b) Major oxides. Fe denotes any one or more entries for feo, feo total, fe2o3, or fe2o3 total. c) Isotope ratios and episilon values



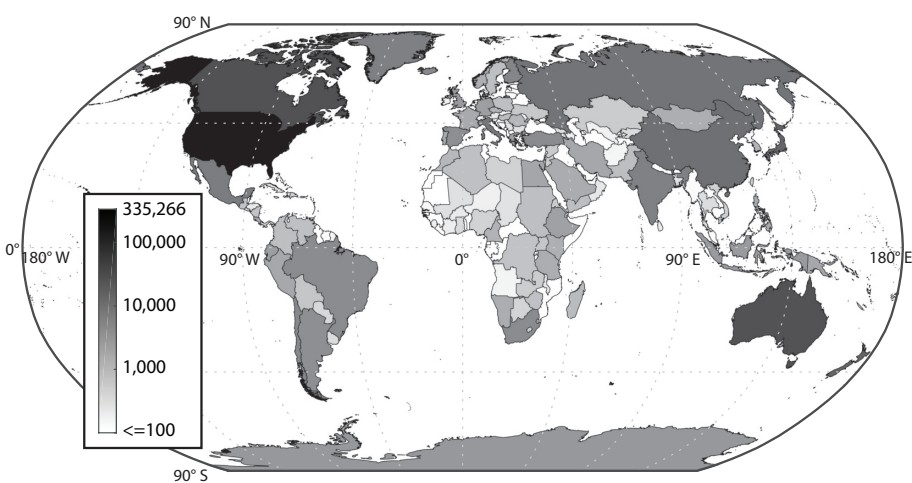

**Figure 3.** Spatial distribution of geochemical samples. Countries are shaded based on the amount of data points within the polygons.





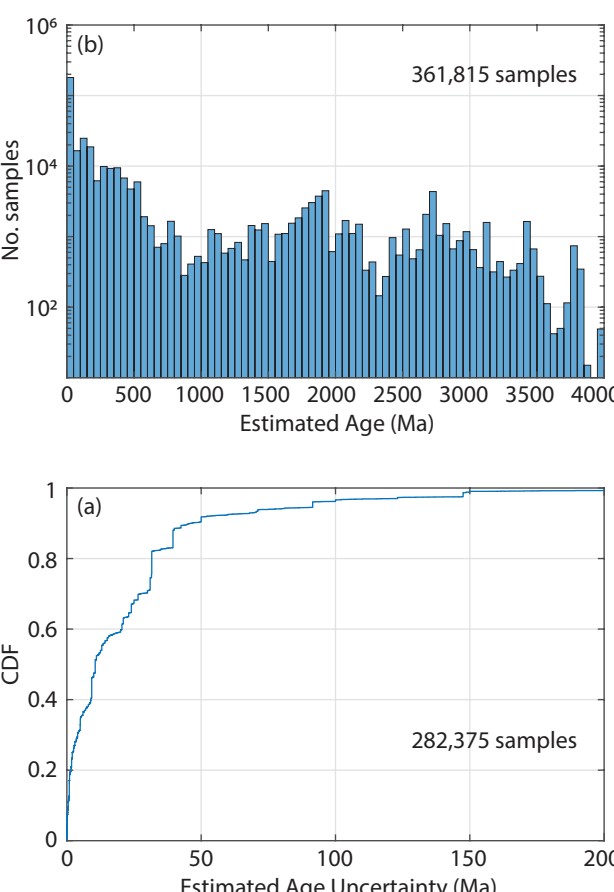

**Figure 4.** Temporal distribution of geochemical samples. a) Histogram of mean ages in 50 Ma intervals b) Empirical CDF of age uncertainty (major time-period associated ages removed)



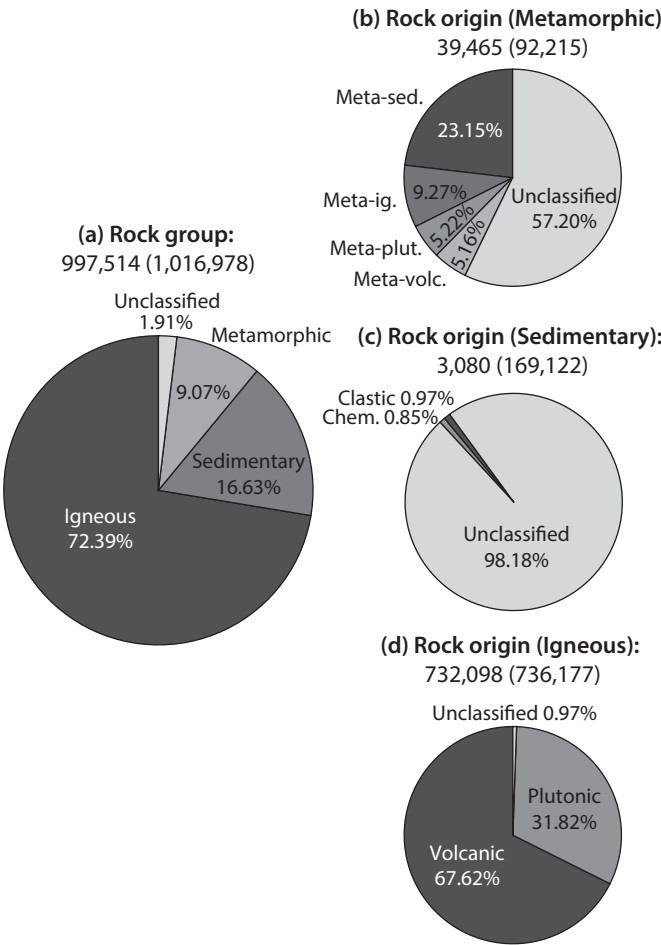

**Figure 5.** Rock group partitioning. a) Pie chart depicting distribution of samples containing a rock group, b) c) and d) denote the rock origin distributions of the rock group fields where rock origin is listed.



**Figure 6.** Rock type classification information. a) Igneous and metaigneous sample histograms of assigned rock names b) Sedimentary and metasedimentary sample histograms of assigned rock names c) TAS igneous classification (Middlemost, 1994) d) High-Mg igneous classification. See Le Bas and Streckeisen (1991) for further information on classification methods. e) Sedimentary classification, after Herron (1988) (Sandclass™)




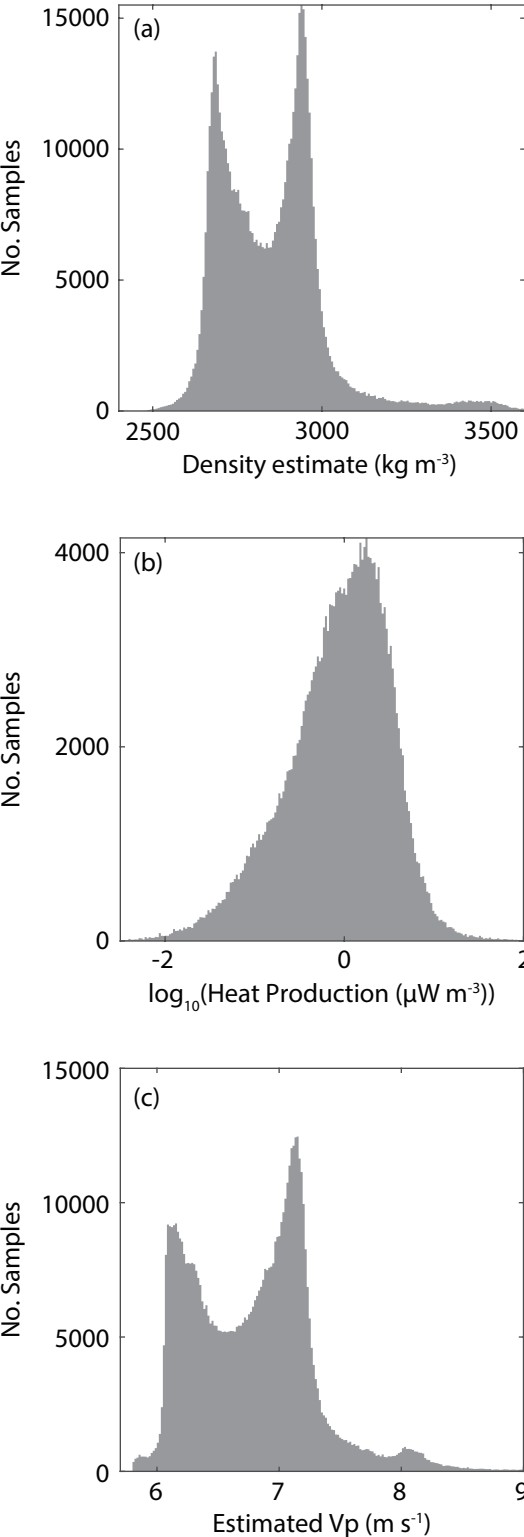

**Figure 7.** Example computed physical property estimate distributions. a) Density b) Heat Production c) P-wave velocity

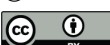



**Table 1.** Brief table content information

| Table name | Table description |
| --- | --- |
| sample | Lists all samples, where sample_id uniquely describes each row. Contains all foreign keys linking to the other tables. Other information such as coordinates, measured density and depth of sample, analysis method, as well as author prescribed sample descriptions, comments and rock names are also included. |
| major | Unique major analyses, linked via the key major_id to sample list. Includes major element oxides as well as volatile, carbonate and l.o.i. content where available. |
| trace | Unique trace element analyses analyses, linked via the key trace_id to sample list. |
| isotope | Unique isotopic ratio analyses, including epsilon values for Hf, Nd and Sr. Linked via the key iso_id to sample list. |
| computed | List of physical properties including heat production and density estimates, and classifications and indices based on schemes such as TAS (Total alkali-Silica) and ASI (aluminum-saturation index). Theses values are computed on a major element normalised (LOI free) version of the associated sample's trace and major compositions and may not match the raw values listed. We preserve the raw data in the database, and methods for normalisation and computed properties are included in the appendices if one wishes to recompute these computed properties and indices with different parameters. comp_id uniquely describes each row and is linked to the sample table. |
| reference | Includes information on the author of the original paper the data was sourced from, and/or reference to database or other previous compilation the data was sourced from e.g. EarthChem. ref_id links the reference table to the sample table. |
| rockgroup | Uniquely links triplets of rock group, rock origin and rock facies to sample table. For definitions of rock group, origin and facies see Table 3. |
| age | Uniquely links sets of age and time period information to sample table |
| country | Unique list of countries (ISO 3166 ALPHA-2 codes) as well as ocean |
| method | Lists unique method strings detailed in previous publications or databases |





**Table 2.** Data sources

| Data source | No. data |
| --- | --- |
| EarthChem family (excluding GEOROC) (https://www.earthchem.org/) | 380,620 |
| GEOROC (http://georoc.mpch-mainz.gwdg.de) | 351,171 |
| OZCHEM (Champion et al., 2016) | 64,462 |
| Petlab (Strong et al., 2016) | 35,950 |
| Petroch (Haus and Pauk, 2010) | 27,388 |
| Newfoundland and Labrador Geoscience Atlas (Newfoundland and Labrador Geological Survey, 2010) | 10,073 |
| The British Columbia Rock Geochemical Database (Lett and Ronning, 2005) | 8,990 |
| Canadian Database of Geochemical Surveys Open File Reports | 8,766 |
| DODAI (Haraguchi et al., 2018) | 6,701 |
| Finnish Geochemical Database (Rasilainen et al., 2007) | 6,543 |
| Ujarassiorit Mineral Hunt (Geological Survey of Greenland, 2011) | 6,078 |
| The Central Andes Geochemical GPS Database (Mamani et al., 2010) | 1,970 |
| Geochemical database of the Virunga Volcanic Province (Barette et al., 2017) | 908 |
| Other sources (∼1,900 sources, misc. files, see reference csv and bib file) | 113,870 |
| Total | 1,023,490 |





**Table 3.** Potentially ambiguous column information

| Column name | Description |
|---|---|
| sample_name | Author denoted title for the sample. Often non-unique e.g. numbered. |
| loc_prec | Location precision |
| qgis_geom | PostGIS ST_Geometry object based on the latitude and longitude of the sample. |
| material | Material/source of the sample e.g. Auger sample, core, drill chips, xenolith, vein |
| rock_name | Rock name designated by the original author |
| sample_description | Sample description mostly inherited from previous databases. Highly variable field. |
| density | Measured density |
| comments | Misc. comments, often additional information not included in the sample description field. |
| method | Method utilised to analyse chemistry and/or age. Variable due to inheritance from previous databases. Multiple methods may be listed, separated by semicolons. |
| norm_factor | Major element normalisation factor applied to the samples major element chemistry before computing properties |
| MALI | the modified alkali–lime index (Frost et al., 2001) |
| fe_number | Iron number (Frost et al., 2001) |
| mg_number | Magnesium number. Fe2+ estimated using $0.85 \times \text{FeO}^T$. |
| asi | Alumina Saturation Index (Frost et al., 2001) |
| maficity | $n_{Fe} + n_{Mg} + n_{Ti}$ |
| cia | Chemical index of alteration (Nesbitt and Young, 1989). Generally CaO* includes an additional correction for $CO_2$ in silicates, but $CO_2$ is not reported for a large fraction of the dataset so we do not include this term for consistency. |
| wip | Weathering Index of Parker (Parker, 1970) |
| spar | Modified from (Debon and Le Fort, 1983) to remove apatite |
| cai | Calcic-alkalic index (Frost et al., 2001) |
| ai | alkalic index (Frost et al., 2001) |
| cpa | Chemical proxy of alteration (Buggle et al., 2011) |
| qtzindex | (Debon and Le Fort, 1983) |
| r1 | R1R2 chemical variation diagram (la Roche et al., 1980) |



| | |
|---|---|
| r2 | R1R2 chemical variation diagram (la Roche et al., 1980) |
| rock_type | compositionally based rock names, discussed in Section 4.2.1, following similar methods of Hasterok et al. (2018) |
| sia_scheme | S-, I-, and A-type granite classification. For felsic compositions, A- and I-types are not properly discriminated with this method. (Frost et al., 2001) |
| frost_class1 | Magnesian or Ferroan (Frost et al., 2001) |
| frost_class2 | Calcic, calc-alkalic, alkali-calcic, alkalic(Frost et al., 2001) |
| frost_class3 | Metaluminous, peraluminous, peralkaline (Frost et al., 2001) |
| quartz | Estimate of quartz content from major element analyses. $SiO_2/M_{SiO_2}$ where $M_X$ is the molecular weight of the oxide $X$ (Mason, 1952; Turekian, 1969) |
| feldspar | Estimate of feldspar/clay/Fe-Al oxide content from major element analyses. $Al_2O_3/M_{Al_2O_3} + Fe_2O_3(t)/M_{Fe_2O_3}$ where $M_X$ is the molecular weight of the oxide $X$ (Mason, 1952; Turekian, 1969) |
| lithics | Estimate of lithics (carbonate) content from major element analyses. $MgO/M_{MgO} + CaO/M_{CaO}$ where $M_X$ is the molecular weight of the oxide $X$ (Mason, 1952; Turekian, 1969) |
| facies | metamorphic facies information pulled from rock_name via partial string search |
| texture | metamorphic texture information pulled from rock_name via partial string search |
| p_velocity | To estimate seismic velocity we use an empirical model developed by Behn and Kelemen (2003), and utilised in Hasterok and Webb (2017). We use the compositional model $V_p(km/s) = 6.9 - 0.011C_{SiO_2} + 0.037C_{MgO} + 0.045C_{CaO}$ where the concentration of each oxide is in wt.%. |
| density_model | We utilise the multiple density estimate methods as outlined by Hasterok et al. (2018) for each compositional group, using multiple linear regression on the data set |
| heat_production_mass | Determined from the chemical composition with the relationship $HP_{mass} = 10^{-5}(9.67C_U + 2.56C_{Th} + 2.89K_2O)$ where C are the concentrations of the HPEs in ppm except $K_2O$ in wt.% (Rybach, 1988) |





| | |
|---|---|
| heat_production | Heat production mass multiplied by the density estimate (in kg $m^{-3}$) (Rybach, 1988) |
| age_ or time_period_ min | Minimum crystallisation age estimate |
| age or time_period | Mean crystallisation age estimate |
| age_ or time_period_ max | Maximum crystallisation age estimate |
| age_sd | Age uncertainty |
| rock_group | The highest order rock type classifications: Igneous/metamorphic/sedimentary |
| rock_origin | Second order classifications of the rock groups - e.g. plutonic/volcanic, metaplutonic/metaigneous/metased, clastic/chemical |
| rock_facies | Third order classifications, mostly restricted to metamorphic rock facies e.g. granulite |
| data_source | Field reserved for existing database compilation e.g. if a sample is derived from EarthChem |
| bibtex | bibtex key corresponding to further reference information if it exists, contained in the attached bib file for easier citation |