# Peer review of "Global whole-rock geochemical database compilation"

_Earth System Science Data, 2019_

## Referee Comment (RC1) · Kent Condie (Referee) · 28 Apr 2019

Excellent contribution to the earth science community, should be widely used. A couple of suggestions to improve the paper:

1) p4, line 10. Indicate isotopic dating method (U/Pb zircon, Sm/Nd garnet, Rb/Sr whole rock, 40Ar/39Ar mica etc). 2) Add uncertainties to isotopic ages 3) Do you include detrital zircon ages? If so they should be reported separately from igneous zircon ages.

p8 Data availability Investigators should be able to download specific parts of the geochemical database, by sorting before download (such as mafic igneous rocks, detrital sediments, tonalities > 2.5 Ga etc.)

[Figure]

Kent Condie

---

## Referee Comment (RC2) · Anonymous Referee #2 · 26 May 2019

Gard and coauthors present a curated database of (primarily) whole-rock elemental and isotopic analyses. This work fills a useful niche between domain-specific manual compilations and large but less-curated online repositories such as EarthChem.

I particularly applaud the authors for ensuring that it is relatively easy to download the full dataset and bibliography in open formats – in this case, csv and bib. There is one minor issue here though that I would request the authors consider addressing: right now, it would not be easy for a user without significant database experience to figure out, e.g., which age corresponds with which sample metadata, or elemental composition, or so on, for any of the ten individual tables provided. The simplest way to address this would be to provide a flat csv of the *entire* dataset. While this would weigh in at perhaps 1 GB, it would be sparse and highly amenable to compression (e.g.,

gzip, for a standard and open option). A similar dataset I have worked with compresses to a relatively manageable 160 MB when treated in such a manner.

Finally (though I suspect this may have already been done) since it is not immediately clear from the text, I would echo the request from Prof. Condie's review, to indicate the isotopic dating method and (critically!) uncertainty for samples with newly-attributed ages.

---

## Referee Comment (RC3) · Juan Carlos Afonso (Referee) · 29 May 2019

This compilation and associated database make up a fantastic contribution to the geoscience community. It should be useful for a wide range of researchers.

The manuscript is well-written and easy to follow. The csv files are clear and easy to download and manipulate. Besides the comments from the other reviewers, the only shortcoming I found at the moment is the lack of interrogation and/or manipulation tools. The authors clearly state that they are creating such tools in matlab, which is terrific, but I would have loved to have at least some basic interrogation codes with this publication! Maybe something the authors can work on for a final version?

Another comment is about the computed properties (Vp, RHP, density). The authors

refer to other works in the text for the methods, which is fine, and then include some equations in Table 3. Can the authors say anything about the uncertainties associated with these estimates? or even better, provide any sort of validation of the predictions against real measurements? I guess that at least some of the samples that made it into the database/s have been characterized well enough to include measured density and perhaps ultrasonic measurements of Vp (?). Such a validation would be great for us readers/users. Perhaps this has been done in the cited works, and if so, all good. I'd then just mention it in the manuscript and give a brief summary to help the reader.

Overall, a really nice contribution. Well done and thanks for your efforts!

---

## Author Comment (AC1) · 7 Jul 2019

Reviewer 1: Kent Condie

We thank Kent Condie for the encouraging comments and suggested revisions.

>"Excellent contribution to the earth science community, should be widely used. A couple of suggestions to improve the paper: 1) p4, line 10. Indicate isotopic dating method (U/Pb zircon, Sm/Nd garnet, Rb/Sr whole rock, 40Ar/39Ar mica etc) 2) Add uncertainties to isotopic ages 3) Do you include detrital zircon ages? If so they should be reported separately from igneous zircon ages."

The 'method' field in the original submission contains the isotopic methods where available, as well as other various methods for geochemical analyses concatenated into a

single entry. Additionally, some information is present in the comments field. This is not ideal, and we agree that separating this information is important. Instead we will separate geochemical method and age dating method into their own fields. Entries are inherited from a vast variety of sources however, and this information was not always supplied or retained, but we will endeavour to expand these fields in each iteration. If a method is supplied, the range of age dates (age_sd, or age_max and age_min range) can be taken to the uncertainty in isotopic age. We do not include detrital zircon ages.

>"Data availability Investigators should be able to download specific parts of the geochemical database, by sorting before download (such as mafic igneous rocks, detrital sediments, tonalities > 2.5 Ga etc.)"

While we understand the Reviewer's desire to have pre-sorted datasets, we philosophically disagree on this point. It is partly this pre-sorting that has led us to design the database in the manner we have. Another reason opted for a single database is we found it annoying to have to download pre-existing databases through web forms in parts because of download limitations. Our dataset is large, but not overly so and smaller than many global datasets. First, pre-sorting requires that several files will be needed to be maintained during subsequent updates. Where does one end with pre-sorting? There are a number of geologically interesting datasets that one might consider e.g. TTGs, komatiites, kimberlites, plume-related magmatism, etc. However, as our understanding of Earth processes grows, our definitions of some of these interesting rocks has evolved. Not to mention the debate surrounding some of these definitions at present. We prefer the database to remain agnostic in this regard. One last point here is that we have developed a set of codes (in Matlab) to parse, filter, plot and run basic computations on the database that is available through github (repository address is included in the text). Not all the codes are available yet because they have not been fully documented, but a complete set of codes is forthcoming. In the future, we may also provide similar codes in Python and/or R.

Reviewer 2: Anonymous reviewer Thank you to the anonymous reviewer for the positive

and constructive comments.

>"Gard and coauthors present a curated database of (primarily) whole-rock elemental and isotopic analyses. This work fills a useful niche between domain-specific manual compilations and large but less-curated online repositories such as EarthChem. I particularly applaud the authors for ensuring that it is relatively easy to download the full dataset and bibliography in open formats – in this case, csv and bib. There is one minor issue here though that I would request the authors consider addressing: right now, it would not be easy for a user without significant database experience to figure out, e.g., which age corresponds with which sample metadata, or elemental composition, or so on, for any of the ten individual tables provided. The simplest way to address this would be to provide a flat csv of the entire dataset. While this would weigh in at perhaps 1 GB, it would be sparse and highly amenable to compression (e.g. gzip, for a standard and open option). A similar dataset I have worked with compresses to a relatively manageable 160 MB when treated in such a manner."

A compressed version of the database in a single, flat file format will be added to the file list as suggested. We agree this may promote easier use of the data for individuals unfamiliar with database structures.

>"Finally (though I suspect this may have already been done) since it is not immediately clear from the text, I would echo the request from Prof. Condie's review, to indicate the isotopic dating method and (critically!) uncertainty for samples with newly-attributed ages."

As addressed in the response to Kent Condie, we will strip the isotopic dating method from the method field where currently retained and endeavour to expand this where information is lacking. Uncertainty has been incorporated through the use of the age_sd, age_max and age_min fields.

Reviewer 3: Juan Carlos Afonso

[Figure]

We thank Juan Carlos Afonso for the positive comments and recommended revisions.

>"This compilation and associated database make up a fantastic contribution to the geo- science community. It should be useful for a wide range of researchers. The manuscript is well-written and easy to follow. The csv files are clear and easy to download and manipulate. Besides the comments from the other reviewers, the only shortcoming I found at the moment is the lack of interrogation and/or manipulation tools. The authors clearly state that they are creating such tools in matlab, which is terrific, but I would have loved to have at least some basic interrogation codes with this publication! Maybe something the authors can work on for a final version?"

We endeavour to have at least a preliminary suite of Matlab scripts available prior to the final publication. Regardless a full suite of scripts will be available later this year, which will be accompanied with documentation and a manuscript of its own.

>"Another comment is about the computed properties (Vp, RHP, density). The authors refer to other works in the text for the methods, which is fine, and then include some equations in Table 3. Can the authors say anything about the uncertainties associated with these estimates? or even better, provide any sort of validation of the predictions against real measurements? I guess that at least some of the samples that made it into the database/s have been characterized well enough to include measured density and perhaps ultrasonic measurements of Vp (?). Such a validation would be great for us readers/users. Perhaps this has been done in the cited works, and if so, all good. I'd then just mention it in the manuscript and give a brief summary to help the reader. Overall, a really nice contribution. Well done and thanks for your efforts!"

The computed properties sections are a little sparse on details of uncertainty and discussion of validity in the current iteration of the manuscript. These details are available in the cited works but should probably be mentioned within this manuscript too. We will include a brief summary of this information as suggested.

[Figure]

2019.

---

## Author Response (AR1)

Dear Editor,

We would like to thank the Editor for giving us the opportunity to respond to the reviewer comments, and the reviewers for their constructive feedback. Concerns and points highlighted by the reviewers are addressed in the revised manuscript. The majority of the edit is incorporated in the addition of an age dating method field, and further discussion about computed properties in Section 4.2. Minor rewrites to tidy sentence structure were also scattered throughout the manuscript.

Kind regards,

Matthew Gard, Derrick Hasterok, Jacqueline A. Halpin

Reviewer 1: Kent Condie

"Excellent contribution to the earth science community, should be widely used. A couple of suggestions to improve the paper: 1) p4, line 10. Indicate isotopic dating method (U/Pb zircon, Sm/Nd garnet, Rb/Sr whole rock, 40Ar/39Ar mica etc) 2) Add uncertainties to isotopic ages 3) Do you include detrital zircon ages? If so they should be reported separately from igneous zircon ages."

Dating method has been parsed out as its own field now (age_method) where available (Figure 1 and Table 3). If a method is supplied, the range of age dates (age_sd, or age_max and age_min range) can be taken to the uncertainty in isotopic age. We do not include detrital zircon ages.

"Data availability Investigators should be able to download specific parts of the geochemical database, by sorting before download (such as mafic igneous rocks, detrital sediments, tonalities > 2.5 Ga etc.)"

While we understand the Reviewer's desire to have pre-sorted datasets, we philosophically disagree on this point. It is partly this pre-sorting that has led us to design the database in the manner we have. Another reason opted for a single database is we found it annoying to have to download pre-existing databases through web forms in parts because of download limitations. Our dataset is large, but not overly so and smaller than many global datasets. First, pre-sorting requires that several files will be needed to be maintained during subsequent updates. Where does one end with pre-sorting? There are a number of geologically interesting datasets that one might consider e.g. TTGs, komatiites, kimberlites, plume-related magmatism, etc. However, as our understanding of Earth processes grows, our definitions of some of these interesting rocks has evolved. Not to mention the debate surrounding some of these definitions at present. We prefer the database to remain agnostic in this regard. One last point here is that we have developed a set of codes (in Matlab) to parse, filter, plot and run basic computations on the database that is available through github (repository address is included in the text). Not all the codes are available yet because they have not been fully documented, but a complete set of codes is forthcoming. In the future, we may also provide similar codes in Python and/or R.

Reviewer 2 - Anonymous Reviewer

"Gard and coauthors present a curated database of (primarily) whole-rock elemental and isotopic analyses. This work fills a useful niche between domain-specific manual compilations and large but less-curated online repositories such as EarthChem. I particularly applaud the authors for ensuring that it is relatively easy to download the full dataset and bibliography in open formats – in this case, csv and bib. There is one minor issue here though that I would request the authors consider addressing: right now, it would not be easy for a user without significant database experience to figure out, e.g., which age corresponds with which sample metadata, or elemental composition, or so on, for any of the ten individual tables provided. The simplest way to address this would be to provide a flat csv of the entire dataset. While this would weigh in at perhaps 1 GB, it would be sparse and highly amenable to compression (e.g. gzip, for a standard and open option). A similar dataset I have worked with compresses to a relatively manageable 160 MB when treated in such a manner."

A complete merged csv is now included and compressed, and does indeed total around 180 MB. Included in the github codes link is also a script to merge desired subtables into a singular table.

"Finally (though I suspect this may have already been done) since it is not immediately clear from the text, I would echo the request from Prof. Condie's review, to indicate the isotopic dating method and (critically!) uncertainty for samples with newly-attributed ages."

As addressed in Condie's response, we now have a column age_method which contains this information.

5   Reviewer 3 - Juan Carlos Afonso

"This compilation and associated database make up a fantastic contribution to the geo- science community. It should be useful for a wide range of researchers. The manuscript is well-written and easy to follow. The csv files are clear and easy to download and manipulate. Besides the comments from the other reviewers, the only shortcoming I found at the moment is the lack of interrogation and/or manipulation tools. The authors clearly state that they are creating such tools in matlab, which is
10   terrific, but I would have loved to have at least some basic interrogation codes with this publication! Maybe something the authors can work on for a final version?"

A number of codes are now available on github (link within text) for managing, filtering and working with the data.

"Another comment is about the computed properties (Vp, RHP, density). The authors refer to other works in the text for the methods, which is fine, and then include some equations in Table 3. Can the authors say anything about the uncertainties
15   associated with these estimates? or even better, provide any sort of validation of the predictions against real measurements? I guess that at least some of the samples that made it into the database/s have been characterized well enough to include measured density and perhaps ultrasonic measurements of Vp (?). Such a validation would be great for us readers/users. Perhaps this has been done in the cited works, and if so, all good. I'd then just mention it in the manuscript and give a brief summary to help the reader. Overall, a really nice contribution. Well done and thanks for your efforts!"

20   The computed properties sections was a little sparse on details of uncertainty and discussion of validity in the original manuscript. These details are available in the cited works as you suggested, but are now also with a brief description in this manuscript in Section 4.2.

We would like to thank all reviewers for their constructive suggestions which aided in improving the manuscript and database, and their positive comments towards our work.

[revised manuscript text omitted]

| Finnish Geochemical Database (**?**) | 6,543 |
| Ujarassiorit Mineral Hunt (**?**) | 6,078 |
| The Central Andes Geochemical GPS Database (**?**) | 1,970 |
| Geochemical database of the Virunga Volcanic Province (**?**) | 908 |
| Other sources (∼1,900 sources, misc. files, see reference csv and bib file) |  123,095 |
| Total | 1, 022,092 |

**Table 3.** Potentially ambiguous column information

| Column name | Description |
| --- | --- |
| sample_name | Author denoted title for the sample. Often non-unique e.g. numbered. |
| loc_prec | Location precision |
| qgis_geom | PostGIS ST_Geometry object based on the latitude and longitude of the sample. |
| material | Material/source of the sample e.g. Auger sample, core, drill chips, xenolith, vein |
| rock_name | Rock name designated by the original author |
| sample_description | Sample description mostly inherited from previous databases. Highly variable field. |
| density | Measured density |
| comments | Misc. comments, often additional information not included in the sample description field. |
| method | Method utilised to analyse chemistry and/or age. Variable due to inheritance from previous databases. Multiple methods may be listed, separated by semicolons. |
| norm_factor | Major element normalisation factor applied to the samples major element chemistry before computing properties |
| MALI | the modified alkali–lime index (**?**) |
| fe_number | Iron number (**?**) |
| mg_number | Magnesium number. Fe2+ estimated using $0.85 \times \mathrm{FeO}^T$. |
| asi | Alumina Saturation Index (**?**) |
| maficity | $n_{Fe} + n_{Mg} + n_{Ti}$ |
| cia | Chemical index of alteration (**?**). Generally CaO* includes an additional correction for $CO_2$ in silicates, but $CO_2$ is not reported for a large fraction of the dataset so we do not include this term for consistency. |
| wip | Weathering Index of Parker (**?**) |
| spar | Modified from (**?**) to remove apatite |
| cai | Calcic-alkalic index (**?**) |
| ai | alkalic index (**?**) |
| cpa | Chemical proxy of alteration (**?**) |
| qtzindex | (**?**) |
| r1 | R1R2 chemical variation diagram (**?**) |
| r2 | R1R2 chemical variation diagram (**?**) |

| | |
|---|---|
| rock_type | compositionally based rock names, discussed in Section 4.1.1, following similar methods of **?** |
| sia_scheme | S-, I-, and A-type granite classification. For felsic compositions, A- and I-types are not properly discriminated with this method. (**?**) |
| frost_class1 | Magnesian or Ferroan (**?**) |
| frost_class2 | Calcic, calc-alkalic, alkali-calcic, alkalic(**?**) |
| frost_class3 | Metaluminous, peraluminous, peralkaline (**?**) |
| quartz | Estimate of quartz content from major element analyses. $SiO_2/M_{SiO_2}$ where $M_X$ is the molecular weight of the oxide $X$ (**??**) |
| feldspar | Estimate of feldspar/clay/Fe-Al oxide content from major element analyses. $Al_2O_3/M_{Al_2O_3} + Fe_2O_3(t)/M_{Fe_2O_3}$ where $M_X$ is the molecular weight of the oxide $X$ (**??**) |
| lithics | Estimate of lithics (carbonate) content from major element analyses. $MgO/M_{MgO} + CaO/M_{CaO}$ where $M_X$ is the molecular weight of the oxide $X$ (**??**) |
| facies | metamorphic facies information pulled from rock_name via partial string search |
| texture | metamorphic texture information pulled from rock_name via partial string search |
| p_velocity | To estimate seismic velocity we use an empirical model developed by **?**, and utilised in **?**. We use the compositional model $V_p(km/s) = 6.9 - 0.011C_{SiO_2} + 0.037C_{MgO} + 0.045C_{CaO}$ where the concentration of each oxide is in wt.%. |
| density_model | We utilise the multiple density estimate methods as outlined by **?** for each compositional group, using multiple linear regression on the data set |
| heat_production_mass | Determined from the chemical composition with the relationship $HP_{mass} = 10^{-5}(9.67C_U + 2.56C_{Th} + 2.89K_2O)$ where C are the concentrations of the HPEs in ppm except K$_2$O in wt.% (**?**) |
| heat_production | Heat production mass multiplied by the density estimate (in kg $m^{-3}$) (**?**) |

345

| | |
|---|---|
| age_ or time_period_ min | Minimum crystallisation age estimate |
| age or time_period | Mean crystallisation age estimate |
| age_ or time_period_ max | Maximum crystallisation age estimate |
| age_sd | Age uncertainty |
| age_method | Method of age estimation, variable due to inheritance from previous databases |
| rock_group | The highest order rock type classifications: Igneous/metamorphic/sedimentary |
| rock_origin | Second order classifications of the rock groups - e.g. plutonic/volcanic, metaplutonic/metaigneous/metased, clastic/chemical |
| rock_facies | Third order classifications, mostly restricted to metamorphic rock facies e.g. granulite |
| data_source | Field reserved for existing database compilation e.g. if a sample is derived from EarthChem |
| bibtex | bibtex key corresponding to further reference information if it exists, contained in the attached bib file for easier citation |